# A Meta-Analysis of the Effects of Insects in Feed on Poultry Growth Performances

**DOI:** 10.3390/ani9050201

**Published:** 2019-04-28

**Authors:** Nassim Moula, Johann Detilleux

**Affiliations:** Fundamental and Applied Research for Animals and Health, University of Liège, 4000 Liège, Belgium; Nassim.Moula@uliege.be

**Keywords:** insects in feed, alternative protein source, poultry growth, meta-analysis

## Abstract

**Simple Summary:**

Today, insects are receiving great attention as a potential source of poultry feed and the number of experiences is exploding. However, it is difficult to obtain an evidence-based view from this large volume of and large diversity of information. A meta-analysis is the best method to summarize the findings of all these studies. Thus, we searched all recent studies that explore the effects of insects in feed on the growth performances of poultry species. Results showed that insects in feed do not modify performances if they substitute less than 10% of conventional protein sources and are not grasshoppers.

**Abstract:**

We investigated and summarized results from studies evaluating the effects of feeding poultry with insects on their growth performances. After a systematic review of studies published since 2000, two independent reviewers assessed the eligibility of each one based on predefined inclusion criteria. We extracted information on the study design, insects, avian species, and growth performances, i.e., average daily gain, feed intake, and feed conversion ratio. Next, we estimated pooled differences between performances of poultry fed a diet with vs. without insects through random-effects meta-analysis models. Additionally, these models evaluated the effects of potential sources of heterogeneity across studies. Of the 75 studies reviewed, 41 met the inclusion criteria and included 174 trials. With respect to diets without insects, pooled differences in growth performances were statistically not different from the null, but heterogeneity was marked across studies. Average daily gain decreased with increasing inclusion rates of insects, going below the null for rates of 10% and more. Grasshoppers were negatively associated with the average daily gain and positively associated with feed intake. The country of publication was another source of heterogeneity across publications. Overall, our results show insects should substitute only partially conventional protein sources and not be grasshoppers to guarantee the appropriate growth of birds.

## 1. Introduction

Today, insects are receiving great attention as a potential source of poultry feed due to the high costs and limited future availability of conventional feed resources, such as soymeal and fishmeal. Insects are a natural part of the poultry diet and feeding them to poultry might improve their welfare. Chitin from the insect’s exoskeleton has been shown to have a positive effect on poultry immune systems, which could reduce the use of antibiotics [1]. Another reason for the interest in insects is their ability to reduce the great quantities of manure, which is becoming a serious environmental problem [2]. As of today, the European Commission has not authorized officially insect-based processed animal proteins as feed for poultry, and the feed ban does not apply to whole insects nor to insect derived fats. It is clear that approval of insect proteins in poultry feed should be reached soon [3].

The nutritional composition of most studied species can be found on the website, Feedipedia [4], and reveals that insects are a rich source of protein, essential amino acids, and fat. Accordingly, researchers started to evaluate the effect of the inclusion of these insects on poultry growth performances. The number of these studies increases every year. For example, a search of the terms ‘insect in animal feed’ in the database, PubMed (done on 13 February 2019), yielded a total of 1292 new publications between 2000 and 2018, with 62 between 2000 and 2002, and 355 between 2016 and 2018. The results of these studies are not always consistent and their power is often not enough to provide evidence of a significant association when it does exist. It is therefore difficult to obtain an evidence-based view from this large volume and large diversity of information. Therefore, it is necessary to summarize and critically analyze these individual studies.

Publications on individual studies may be summarized in various forms [5], but one method, meta-analysis (MA), has several advantages over the others. It is less prone to subjective interpretation because it applies objective formulas to summarize findings. It increases the sample size and the power of measuring a potential effect as it combines the results of numerous studies. Also, it can be used with any number of studies. However, several critical issues need to be addressed [6]. For example, one should be aware that publication bias (i.e., studies with statistically significant results are more likely to be published than others) and selection bias (i.e., studies included in the review process) may alter the results of an MA. Also, the homogeneity of findings should be checked as the conclusions of an MA will be less clear if the included studies have differing results. Fortunately, a variety of graphs (e.g., forest, funnel, or Galbraith plots), statistics (e.g., H^2^ or *I*^2^), and random-effects models (e.g., meta-regression) are available to detect heterogeneity and determine whether it is due to one or more characteristics of the studies included in the MA [7,8,9,10]. 

Therefore, our objective is to perform an MA of studies examining the effect of the inclusion of insects on poultry growth performances by using random-effects models.

## 2. Materials and Methods 

To reach our objective, we conducted a systematic search of the literature in Pubmed, Medline, and Google Scholar. We used a combination of keywords and subject headings for the following concepts: Insect, avian species, and feed. A total of 75 articles, published between 2000 and 2019, were independently read/reviewed by both authors. 

We included studies that provide information on the effect of insects (e.g., black soldier fly larvae, house fly maggots, mealworm, locusts, grasshoppers, crickets, silkworm, or caterpillar) under any forms (e.g., fresh, congelated, or dried; ground; or whole) in the feed of any avian species (e.g., poultry, turkey, quail). Diets must be iso-nitrogenous and iso-energetic. Studies must contain information concerning the inclusion rate of insects (from 0% to 100%) as the replacement of conventional sources of protein, a measure of the effect of the diet on the average daily gain (ADG; gr), feed intake (FI; g/day), and/or feed conversion ratio (FCR), and a measure of the variability associated to the effect. The measure of variability could be a standard deviation, standard error of the mean, confidence interval, or mean square error. They were all expressed as standard deviation, after transformation if necessary. Manuscripts had to be original research (not a review or conference abstract), and be written in English or French. For each study, we computed the differences between the means of the ADG, FI, and FCR for poultry fed a diet with vs. without insects (at various rates of substitution). These differences were denoted as DIFF_ADG, DIFF_FI, and DIFF_FCR, respectively. Their corresponding standard deviations had to be in the range of 0.001 to 20.

We implemented two random-effects models in the MA. The first one is the full-model:
y_i_ = μ + t_i_ + e_i_,(1)
where y_i_ is the estimated measure (DIFF_ADG, DIFF_FI, or DIFF_FCR) for the ith trial (i = 1, 2, …, N), N is the number of trials in the MA, μ is the overall mean, and t_i_ and e_i_ are random effects. The t_i_ are assumed to be independent normal variables with a zero mean and between-study variance, v^t^_i_. The e_i_ are assumed to be independent normal variables with a zero mean and within-study variance, v^e^_i_. Heterogeneity across studies was quantified by the index, *I*^2^, i.e., the percent of the total variation due to variation across studies [4]. The second model include the same effects as the first one plus effects for potential sources of heterogeneity across studies:
y_ijkl_ = μ + t_i_ + h_ij_ + s_ik_ + c_il_ + b_1_p_ijkl_ + b_2_ a_ijkl_ + e_ijkl_,(2)
where y_ijkl_ is the measure for the ith trial (i = 1, 2, …, N), jth animal category j (j = 1, 2, 3), kth insect species (k = 1, 2, ..., 5), and lth continent where the study was carried out (l = 1, 2, 3, 4). Fixed effects are h_ij_ for the categories of animal species (i.e., broilers, layers, and others), s_ik_ for the categories of insect species (i.e., black soldier fly larvae, house fly maggots, mealworms, grasshoppers, and others), c_il_ for the continent in which the study was carried out (i.e., Europe, Africa, Americas, Asia-Oceania), p_ijkl_ for the percent of insects’ inclusion (from 0% to 100%), and a_ijkl_ for the year of publication (from 2000 to 2019). The parameters, b_1_ and b_2_, are the regression coefficients relating the inclusion rate and the year of publication to the measure, respectively. The amount of heterogeneity that is accounted for by the effects included in the model is given by the pseudo-R^2^ value [11]. We used the function “rma” of the package “bayesmeta” to fit the models to the data, obtain estimates of the effects included in both models, and create forest and funnel plots [12]. The *p* value threshold for statistical significance was set at 1%.

## 3. Results

After deduplication and screening for inclusion criteria, 41 studies [2,13,14,15,16,17,18,19,20,21,22,23,24,25,26,27,28,29,30,31,32,33,34,35,36,37,38,39,40,41,42,43,44,45,46,47,48,49,50,51] and 174 trials were selected for the MA (Appendix A). Insects mostly represented were black soldier fly larvae (29.89%), mealworms (20.11%), maggots (14.37%), grasshoppers (12.64%), and others (22.99%), such as crickets, silkworms, or locusts. Typically, insects were provided as a dried and ground (defatted or not) meal obtained from specialized companies. Birds were mostly broilers (68.39%) and laying hens (13.22%). Other birds (18.39%) included quails, guinea fowls, or partridges. In 28% of the trials, insects substituted less than 10% of conventional protein sources (Figure 1).

Studies in the MA were mostly from African and European countries and their numbers increased with the year of their publication (Figure 2). 

In Figure 3 and Figure 4, one can find forest and funnel plots for DIFF_ADG, DIFF_FI, and DIFF_FCR, respectively. Forest plots illustrate that most individual 95% confidence intervals (CI) include the null value and do not perfectly overlap, which suggests heterogeneity between studies. Funnel plots point to a broad absence of publication biases. 

Pooled estimates (and their 95% CI) and the percent of total variation across studies due to heterogeneity are given in Table 1 (results from the first model). Pooled estimates are statistically not different from the null for DIFF_ADG, DIFF_FI, and DIFF_FCR. However, values of *I*^2^ suggest strong heterogeneity across studies for all differences. 

Results of the analysis of the potential causes of this heterogeneity are given in Table 2 (results of the second model). One striking observation is that DIFF_ADG decreased significantly as the percent of insects included in the diet increased: It decreased by 0.05 g for each percent increase in dietary insects. This finding is also illustrated in Figure 5. From Table 2, one can estimate that the ADG of birds fed on a diet with insects is significantly lower than the ADG of birds fed a diet without insects once inclusion rates are 10% and more. 

Another finding is that DIFF_FI is the lowest for birds eating maggots. For those eating grasshoppers, DIFF_FI is the highest and DIFF_ADG is the lowest. More precisely, the FI of birds eating grasshoppers is 3.83 g higher than the FI of birds eating black soldier flies and the ADG of birds eating grasshoppers is 4.32 g less than the ADG of birds eating black soldier flies. 

Absolute values of DIFF_ADG, DIFF_FI, and DIFF_FCR were the highest in studies published in Asia-Oceania. Also, DIFF_ADG is lowest for animals other than broilers or laying hens. Finally, given the R^2^ values, one may expect that sources of heterogeneity other than the ones considered in this study exist.

## 4. Discussion

In this study, we sought to evaluate in an MA the effects of dietary insects on poultry performances from recently published studies. Results of the MA showed that the inclusion of insects had no statistically significant overall adverse effect on the ADG, FI, and FCR. This confirms findings in other review studies (e.g., [45]), but does not take into account the heterogeneity across studies as revealed by the large values of *I*^2^, especially for DIFF_ADG and DIFF_FCR (Table 1).

Indeed, increasing rates of insect inclusion are associated with a decrease of ADG (Table 2) in birds, especially for rates of 10% and more. Although diets in this MA were iso-nitrogenous and iso-energetic, this observation could be associated with an imbalance in the nutrient profile, albeit amino acids profiles in black soldier fly larvae, maggots, and mealworms seem ideal for broilers [4]. Another hypothesis could be that chitin in high amounts is less digestible. However, Hossain and Blair [52] showed that the introduction, up to 100%, of commercial chitin derived from crustacean shell waste in the diet of broilers had no statistically significant effects on their ADG and FI. Similarly, Tabata et al. [53] reported chicken stomach tissues express high levels of acidic chitinase mRNA and their translation products can degrade chitin in the gastro-intestinal tract. 

Whatever the etiology for the decrease in ADG, it is supported by the observed effect of insects feeding on the morphology of intestinal villi. Indeed, a decrease in intestinal villi heights has been observed in laying hens fed high levels of black soldier fly larvae [54] and in Ross fed high levels of mealworms [20], but not in Ross fed low levels of black soldier fly larvae [41] nor in free-range chickens fed low levels of mealworms [55]. By shortening villi, the total luminal villus absorptive area is decreased together with the nutrient metabolizability and performance [56]. Insect feeding could also modify the intestinal microbiota as it was suggested in a study of Label Hubbard chickens fed mealworms [55], but not in a study of Ross fed black soldier fly larvae [41]. These last results are speculative because many host-related and environmental factors have a large effect on the composition of intestinal microbiota [57]. 

In our study, it was also observed that birds eating grasshoppers lose weight when compared with those eating black soldier fly larvae. This may be related to the poor amino-acid profile in grasshoppers and the low digestibility of their crude protein fraction [4]. Finally, the observation that the ADG is lowest for birds other than broilers is not surprising because this category includes quails, guinea fowls, or partridges that have not been subjected to as intense a selection for performance as broilers.

Here, birds eating maggots have a were more likely to eat less than those eating black soldier fly larvae. Inversely, birds eating grasshoppers were more likely to eat more than those eating black soldier fly larvae. One tentative explanation is that birds have a tendency to eat larger particles (e.g., [58]). Indeed, grasshoppers and black soldier fly larvae are generally larger than maggots. However, most insects in the MA were provided as a dried and ground meal. Another explanation may be that the texture and color of the feed containing maggots render the feed less palatable, and inversely for grasshoppers. 

No effect in this MA could explain the differences in the FCR across studies, with the exception of the continent where the study was carried out. Methodological issues and management factors may explain that estimates of DIFF_ADG, DIFF_FI, and DIFF_FCR were all better in Asia-Oceania than in studies carried out in Europe. Indeed, R^2^ values (Table 2) suggest that sources of heterogeneity exist across studies, other than the ones assessed in this MA. Management factors include characteristics of the local environment (e.g., temperature and ventilation), age and sex of the birds, quality of the diet nutrient (e.g., quality of amino acids), or the structure (e.g., ground or not) and stage (e.g., larvae or adult) of the insects. Methodological issues include measures of variation used in reporting mean effects (i.e., standard error of the mean, confidence interval, or mean square error), methods of computation (e.g., FCR can be computed from measures of the ADG and FI and vice versa), technologies and instruments to evaluate nutrient composition and growth performances (e.g., exactitude of the scales), or the level of physical activity of the birds (e.g., restricted or open area). This is a caveat of this MA and we could not find any quality criteria checklist, such as those proposed to evaluate the quality of studies that evaluate health care interventions (e.g., [59]).

## 5. Conclusions

Insects are suggested to be included in poultry feed and the number of experiences is exploding. However, it is difficult to obtain an evidence-based view of their effects on poultry performances from this large volume and large diversity of information. This MA allowed us to formally and systematically pool together all relevant research and clarify findings based on all currently available information. This is important for authorities to make decisions about the approval of the inclusion of insect protein in poultry feed. 

Overall, the results of the MA showed that insects should substitute only partially conventional protein sources and should not be grasshoppers to guarantee the appropriate growth of birds. In such cases, the inclusion of insects had no overall adverse effect on the ADG, FI, and FCR. This conclusion applies to the insects (i.e., mostly black soldier fly larvae, mealworms, and maggots) and poultry species (i.e., mostly broilers) represented in the MA and cannot be generalized to others. Results also pointed to the presence of heterogeneity across findings in studies that evaluate the effects of insects in feed on animal performances and the need for a checklist to evaluate the quality of such studies.

## Figures and Tables

**Figure 1 animals-09-00201-f001:**
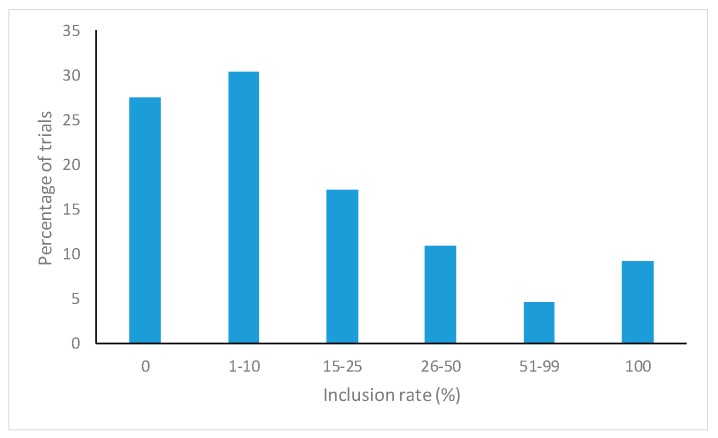
Inclusion rates of insects in the diet of poultry in the trials included in the meta-analysis.

**Figure 2 animals-09-00201-f002:**
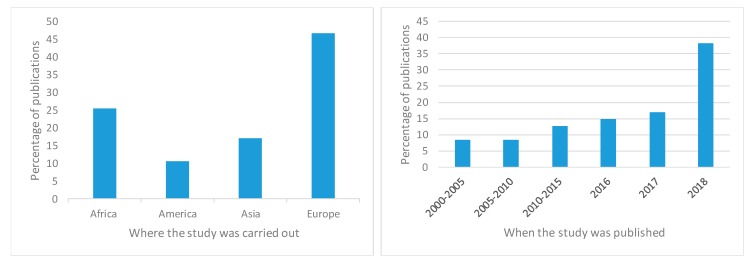
Repartition of studies included in the meta-analysis per continent and per year of publication.

**Figure 3 animals-09-00201-f003:**
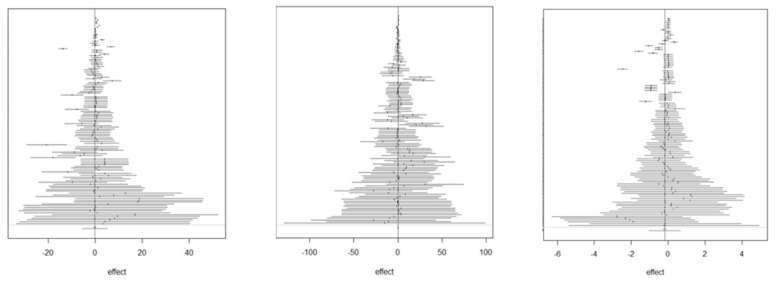
Forest plots of the differences in means of the average daily gain (left panel), feed intake (middle panel), and feed conversion ratio (right panel) between poultry fed a diet with and without insects.

**Figure 4 animals-09-00201-f004:**
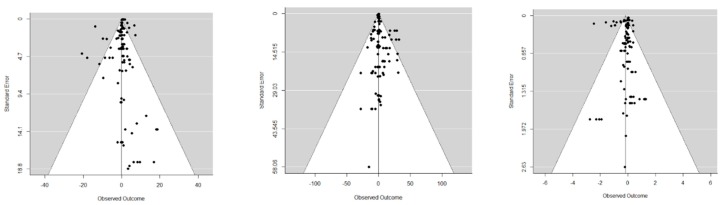
Funnel plots of the differences in means of the average daily gain (left panel), feed intake (middle panel), and feed conversion ratio (right panel) between poultry fed a diet with and without insects.

**Figure 5 animals-09-00201-f005:**
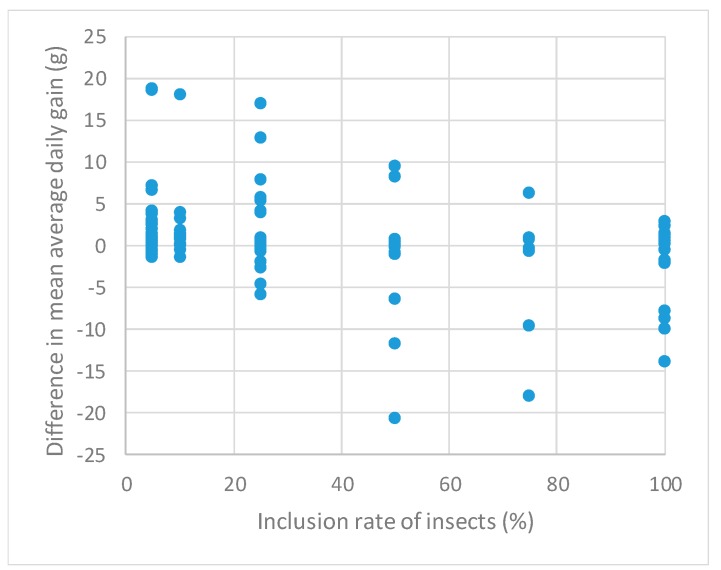
Differences in means of the average daily gain between poultry fed a diet with vs. without insects per rate of their inclusion in the diet.

**Table 1 animals-09-00201-t001:** Results of the full model of analysis without accounting for sources of heterogeneity.

Differences between Poultry Fed a Diet With vs. without Insects in Means of	Pooled Estimate	Heterogeneity (*I*^2^)
Average daily gain	−0.10 (−0.83 to 0.63)	99.2
Feed intake	0.14 (−0.18 to 0.41)	39.9
Feed conversion ratio	−0.18 (−0.29 to −0.07)	89.6

**Table 2 animals-09-00201-t002:** Estimates of the effects of the characteristics of the study (continent and year of publication) and of the trial (categories of birds, of insects, and percent of insects’ inclusion) on the differences in means of the average daily gain (DIFF_ADG), feed intake (DIFF_FI), and feed conversion ratio (DIFF_FCR) between poultry fed a diet with vs. without insects.

Effects	DIFF_ADG (g)	DIFF_FI (g)	DIFF_FCR
Overall mean	−4.56 (−9.50 to 0.38)	3.77 (−0.83 to 8.42)	0.23 (−0.40 to 0.86)
*Insects species*			
Black soldier fly larvae (reference)	0	0	0
Maggots	3.13 (−1.05 to 7.31)	−6.56 * (−10.87 to −2.26)	−0.02 (−0.48 to 0.45)
Mealworms	1.31 (−0.79 to 3.42)	−1.12 (−2.11 to −0.13)	0.12 (−0.18 to 0.42)
Grasshoppers	−4.32 * (−6.83 to −1.81)	3.83 * (1.43 to 6.24)	−0.21 (−0.67 to 0.24)
Other insects	−0.77 (−2.69 to 1.15)	−1.49 (−3.45 to 0.48)	−0.10 (−0.34 to 0.13)
*Animal species*			
Broilers (reference)	0	0	0
Layers	2.31 (0.04 to 4.57)	−1.25 (−5.66 to 3.15)	−0.05 (−0.88 to 0.17)
Other poultry	1.42 (−0.31 to 3.15)	−4.41 (−6.60 to −2.21)	−0.12 (−0.47 to 0.22)
*Inclusion rate*	−0.05 * (−0.08 to −0.03)	−0.005 (−0.02 to 0.01)	−0.003 (−0.006 to −0.001)
*Year of publication*	0.29 (0.04 to 0.54)	0.044 (−0.19 to 0.28)	−0.007 (−0.039 to 0.024)
*Continent*			
Europe (reference)	0	0	0
Africa	0.58 (−1.15 to 2.31)	1.44 (−0.91 to 3.80)	0.14 (−0.17 to 0.45)
Asia and Oceania	4.46 * (2.22 to 6.70)	−3.41 * (−5.70 to −1.13)	−0.50 * (−0.78 to −0.22)
America	−0.05 (−3.88 to 3.78)	−1.87 (−5.96 to 2.22)	−0.22 (−0.73 to 0.29)
Amount of heterogeneity accounted for (R^2^, %)	52.82	76.93	47.37

* *p* value < 0.001.

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
