# Peer review of "A Meta-Analysis of the Effects of Insects in Feed on Poultry Growth Performances"

_animals, 2019, doi:10.3390/ani9050201_

Round 1

Reviewer 1 Report

The manuscript summarizes the results obtained by a wide bibliographic research applying a statistical R package. The criteria of evaluation are generally consistent and it is clear that the methods have been applied by researchers with good skills in the use of statistical tools.

 Nevertheless, even if the results are of good quality, the manuscript results are not very approachable for journal readers and different details are considered as deductive. 

Some perplexity:

- the mathematical formulas (first and second method applied) have already been used by other authors or are equations from R'rma' package? in the first case, mention them, in the second case you shoul better explain since you never speack about R'rma' in the main text.

- line 99. the p-value threshold (1%) for significance is referred to FDR (false discovery rate)? in this case you shoul specify it.

- fig. 1. you are saying that only about 27% of studies considered used 0% insect meal inclusion (first column). This makes about the 70% of studies considered not consistent with the others

- line 121. What is CI? it is never spelled out in the text

- line 186-187. maggots (grasshoppers), less (more). What do you mean?

Some comment:

- You should better explain the meaning of each variable used in the formulas used. Methods are too simplificative. It should be usefull to provide more explanations and a better understandeble list of abbreviations for each formula, at least for the first method. Variables should be listed in order of appearance. It is very difficult to read.

- Change layer with laying hens

- The results are descrived in a reductive manner considering the variables in terms of species (birds and insects) and form  of insect meal (fresh, frosted, etc),  

- It is really difficult to interpret plots in particular the "forest" ones since the y axis is not readeble.

- You should extend discussions and write them in a more impersonal manner. Do not use i.e. "one may argue...", "Note however..", 

It is true that insects are receiving growning attention as feed for animals as they are also used, in aquaculture field and in basic science using experimental models whit several implications. You should add in the introduction mentioning reviews or recent studies (such as DOI10.1089/zeb.2018.1596, Zarantoniello et al., 2018; DOI:10.1089/zeb.2017.1559, Vargas et al., 2017)

The manuscript should be presented in a manner that better match the attitudes of journal readers that are not strictly statistical (zootechnical, biologists, veterinarians etc).

I would suggest to consider it after major revision.

Author Response

Thank you for your revison. Please find the responses below

Some perplexity:

- the mathematical formulas (first and second method applied) have already been used by other authors or are equations from R'rma' package? in the first case, mention them, in the second case you shoul better explain since you never speack about R'rma' in the main text.

RESP/ The formulas are specific to the data used in our study. We used the function “rma” of the package “bayesmeta” to fit the models to the data, obtain estimates of the effects included in both models and create forest and funnel plots. Modifications in the paper at L105-107

- line 99. the p-value threshold (1%) for significance is referred to FDR (false discovery rate)? in this case you shoul specify it.

RESP/ In my understanding, the p-value is the probability of a false positive on a single test (the error of rejecting a null hypothesis when it is actually true). Traditionally, it is set as .05 or .01 - as in there is only a 5 or 1 in 100 chance that the variation that we are seeing is due to chance. This is called the 'level of statistical significance' (L 99 in the tex). I think the term FDR is used for large-scale multiple testing which is not the case in this study. Don’t you agree?

- fig. 1. you are saying that only about 27% of studies considered used 0% insect meal inclusion (first column). This makes about the 70% of studies considered not consistent with the others

REP/ Thanks for your remark. I meant ‘trials’, not ‘studies’ (L118). In all studies, diets containing insects were compared to a control without any insects. 

- line 121. What is CI? it is never spelled out in the text

REP/ Thanks for your remark. CI is for confidence interval (L129).

- line 186-187. maggots (grasshoppers), less (more). What do you mean?

REP/ Thanks for your remark. Considering DIFF_FI, results of the MA revealed birds eating maggots have a tendency to eat less than those eating black soldier fly larvae. Inversely, birds eating grasshoppers have a tendency to eat more than those eating black soldier fly larvae. Modifications in the paper at L191-193.

Some comment:

- You should better explain the meaning of each variable used in the formulas used. Methods are too simplificative. It should be usefull to provide more explanations and a better understandeble list of abbreviations for each formula, at least for the first method. Variables should be listed in order of appearance. It is very difficult to read.

REP/I don’t know how to make the methods clearer. Each variable is described from L73-L84. The symbols of the models are classically those employed in meta-analyses.

- Change layer with laying hens

REP/ Done in all the paper

- The results are descrived in a reductive manner considering the variables in terms of species (birds and insects) and form  of insect meal (fresh, frosted, etc),  

REP/I don’t understand the comment. The variables that are discussed are those included in the models of the meta-analysis. We try to extend this part of the paper (L151-164)

- It is really difficult to interpret plots in particular the "forest" ones since the y axis is not readeble.

REP/ Graduations of the Y axis were removed because they are useless. Each line in the plot corresponds to the confidence interval for a trial.

- You should extend discussions and write them in a more impersonal manner. Do not use i.e. "one may argue...", "Note however..", 

It is true that insects are receiving growning attention as feed for animals as they are also used, in aquaculture field and in basic science using experimental models whit several implications. You should add in the introduction mentioning reviews or recent studies (such as DOI10.1089/zeb.2018.1596, Zarantoniello et al., 2018; DOI:10.1089/zeb.2017.1559, Vargas et al., 2017)

REP/ The topics being focused on using insects in poultry feed, we extend discussions with papers on intestinal morphometry and microbiota in poultry (L196-205)? We also extend the introduction by adding some information regarding the potentiality of the insect in poultry nutrition and some information regarding the regulation of the use of insect in Europe (as suggested by the second reviewer) L36-42

The manuscript should be presented in a manner that better match the attitudes of journal readers that are not strictly statistical (zootechnical, biologists, veterinarians etc).

REP/Hopefully, you’ll find it more adequate.

I would suggest to consider it after major revision.

Reviewer 2 Report

In my opinion the Article 474732 titled “A meta-analysis of the effects of insects in feed on poultry growth performances" is an original review in the Nutrition area of Animals. However, I think that this manuscript needs minor revisions before publication.

Objectives: in the manuscript were reported the results of studies evaluating the effects of insects feeding on poultry growth performances.

Abstract: well done.

Keywords: some changes.

Introduction: The introduction must be improved.

Material and methods: The experimental design is clear and well structured.

Results and Discussion: very consistent. 

Conclusion: Must be improved.

References: some references are not reported accurately.

I hope I have contributed to the improvement of the manuscript. 

Detailed Comments

Keywords: add “alternative protein source”

Introduction: this section must be extend adding some information regarding the potentiality of the insect in poultry nutrition and some information regarding the regulation of the use of insect in Europe.

Line 25: please change “gains” in gain.

I can suggest to cite also another new research paper, published in Animals in a special issue, Insects: Alternative Protein Source for Animal Feed, entitled Intestinal Morphometry, Enzymatic and Microbial Activity in Laying Hens Fed Different Levels of a Hermetia illucens Larvae Meal and Toxic Elements Content of the Insect Meal and Diets by Moniello et al. 

Figure 5: please add a secondary axis in this graph in order to better understand the different units (gr and %).

Unit of measure: please change gr in g in all the paper.

References: please check the format of the references according to the guidelines of the Journal. 

Conclusion: this section must be improved, in particular the lines 165-170 of the discussion can be reported in the conclusion.

Author Response

Thank you for your revison. Pleas find below the answers to your comments.

In my opinion the Article 474732 titled “A meta-analysis of the effects of insects in feed on poultry growth performances" is an original review in the Nutrition area of Animals. However, I think that this manuscript needs minor revisions before publication.

Objectives: in the manuscript were reported the results of studies evaluating the effects of insects feeding on poultry growth performances.

Abstract: well done.

Keywords: some changes.

Introduction: The introduction must be improved.

Material and methods: The experimental design is clear and well structured.

Results and Discussion: very consistent. 

Conclusion: Must be improved.

References: some references are not reported accurately.

I hope I have contributed to the improvement of the manuscript. 

Detailed Comments

Keywords: add “alternative protein source”

REP/ Done. Good idea.

Introduction: this section must be extend adding some information regarding the potentiality of the insect in poultry nutrition and some information regarding the regulation of the use of insect in Europe.

REP/ Done. L 36-42

Line 25: please change “gains” in gain.

REP/Done. Thanks. L26.

I can suggest to cite also another new research paper, published in Animals in a special issue, Insects: Alternative Protein Source for Animal Feed, entitled Intestinal Morphometry, Enzymatic and Microbial Activity in Laying Hens Fed Different Levels of a Hermetia illucens Larvae Meal and Toxic Elements Content of the Insect Meal and Diets by Moniello et al. 

REP/I am more than happy to cite this work. L198 

Figure 5: please add a secondary axis in this graph in order to better understand the different units (gr and %).

REP/Done

Unit of measure: please change gr in g in all the paper.

REP/Done

References: please check the format of the references according to the guidelines of the Journal. 

REP/The new version is hopefully better

Conclusion: this section must be improved, in particular the lines 165-170 of the discussion can be reported in the conclusion.

REP/Hopefully the conclusion is improved.

Round 2

Reviewer 1 Report

You moderately answered to my concerns and partially accepted suggestions. As you assert in your conclusion, this work can be usefull to give an overall vision on the use of insect meal in poultry feed giving general informations mainly at decision-making level.